# *OsLPR5* Encoding Ferroxidase Positively Regulates the Tolerance to Salt Stress in Rice

**DOI:** 10.3390/ijms24098115

**Published:** 2023-04-30

**Authors:** Juan Zhao, Xin Meng, Zhaonian Zhang, Mei Wang, Fanhao Nie, Qingpo Liu

**Affiliations:** 1The Key Laboratory for Quality Improvement of Agricultural Products of Zhejiang Province, College of Advanced Agricultural Sciences, Zhejiang A&F University, Hangzhou 311300, China; zhaojuan521321@163.com (J.Z.); mx42032@stu.zafu.edu.cn (X.M.); zjnlnxy193@163.com (Z.Z.);; 2Institute of Horticulture, Zhejiang Academy of Agricultural Sciences, Hangzhou 310021, China

**Keywords:** rice, salinity, *OsLPR5*, ferroxidase, stress

## Abstract

Salinity is a major abiotic stress that harms rice growth and productivity. Low phosphate roots (LPRs) play a central role in Pi deficiency-mediated inhibition of primary root growth and have ferroxidase activity. However, the function of LPRs in salt stress response and tolerance in plants remains largely unknown. Here, we reported that the *OsLPR5* was induced by NaCl stress and positively regulates the tolerance to salt stress in rice. Under NaCl stress, overexpression of *OsLPR5* led to increased ferroxidase activity, more green leaves, higher levels of chlorophyll and lower MDA contents compared with the WT. In addition, OsLPR5 could promote the accumulation of cell osmotic adjustment substances and promote ROS-scavenging enzyme activities. Conversely, the mutant *lpr5* had a lower ferroxidase activity and suffered severe damage under salt stress. Moreover, knock out of *OsLPR5* caused excessive Na^+^ levels and Na^+^/K^+^ ratios. Taken together, our results exemplify a new molecular link between ferroxidase and salt stress tolerance in rice.

## 1. Introduction

Soil salinity is a global environmental problem and an important abiotic stress factor that limits the germination, growth and productivity of plants [1]. High concentrations of salt affect various plant physiological and biochemical processes, causing ionic toxicity, osmotic stress, ROS accumulation and nutritional imbalance [2]. Rice is one of the principal cereal crops for the world’s population and is salt-sensitive in the seedling and reproductive stages [3]. To ensure food security, we need to combat the rising threat of soil salinity and develop and grow grain output under saline conditions [4,5]. However, the research on salt tolerance started relatively late in rice compared with in Arabidopsis, and although numerous salt-responsive genes have been identified, very few have been successfully applied in rice [6,7]. Genetic studies aimed at identifying genes underlying salt tolerance and elucidating the corresponding mechanisms are becoming more urgent [8].

Plants have evolved various systems to sense and adapt to a high salinity environment. With salt stress, plants absorb excess Na^+^ from soil solution into their roots, which moves sequentially to the shoots and throughout the leaves, ultimately inhibiting the absorption of other nutrients such as K^+^ [9,10]. To maintain cellular Na^+^/K^+^ homeostasis, a series of Na^+^/H^+^ antiporter, high-affinity K^+^ transporters (HKT) and Cl^-^ channels help the Na^+^ efflux, uptake restriction and compartmentalize into the vacuole [2,11,12,13]. Proline biosynthesis-related genes such as *OsP5CS* and *OsP5CR* and trehalose-6-phosphate synthase/phosphatase-related genes such as *OsTPP1* and *OsTPS1* can promote the accumulation of osmolytes and protect rice from osmotic stress under saline conditions in rice [14,15]. In addition, plants have enzymatic scavengers and nonenzymatic antioxidants to mitigate oxidative stress caused by ROS accumulation. Enzymatic scavengers include superoxide dismutase (SOD), ascorbate peroxidase (APX), catalase (CAT), glutathione peroxidases (GPXs), glutathione peroxidase (GR), glutaredoxin (GRX), glutathione S-transferase (GST) and respiratory burst oxidase homologs (RBOHs) [6,16,17]. Ascorbic acid, glutathione, carotenoids, flavonoids, alkaloids, phenolic compounds and tocopherol are nonenzymatic antioxidants [1]. In addition to the above physiological mechanisms, there are many functional proteins, transcription factors, hormones, etc., involved in signaling and resisting salt stress.

Low phosphate roots (LPRs) encode the multicopper oxidase domain-containing protein, named for its function in Pi deficiency-mediated inhibition of primary root growth [18]. In Arabidopsis, LPR1 interacts genetically with the P5-type ATPase PDR2 in the endoplamic reticulum (ER), and it plays an opposite role in mediating root meristem growth responses to Pi and Fe availability [19]. Furthermore, *LPR1* encodes a cell-wall-targeted ferroxidase and function in iron-dependent callose deposition in low Pi conditions [20]. There are five homologs of LPR1 in rice (OsLPR1-5), and significant increases in the relative expression levels of OsLPR3 and OsLPR5 could be triggered under Pi deficiency [21]. Similar to Arabidopsis, it was reported that OsLPR5 was located in the ER and the cell wall, had ferroxidase activity and was required for normal growth and maintenance of phosphate homeostasis in rice [22]. Moreover, the concentration of Fe (III) and total Fe were increased in the roots and shoots of OsLPR5-overexpressing plants [22]. Therefore, OsLPR5 has a broad spectrum influence on Fe homeostasis and plant development. However, little is known about LPRs functions in abiotic stress response and tolerance.

Although the important roles of OsLPR5 have been reported in plants, its function in salt stress response and tolerance has yet to be defined. In this study, we identified the expression level of OsLPR5 and ferroxidase activity under NaCl stress. The physiological roles in salt stress and tolerance were further identified through overexpression and CRISPR/Cas9-mediated mutation of *OsLPR5*. We have uncovered a previously unknown role that *OsLPR5* positively regulates the salt stress tolerance in rice.

## 2. Results

### 2.1. OsLPR5 Is Induced under Salt Stress and Exhibits Higher Expression in Vegetative Organs

To determine the effect of salt stress on the relative transcriptional level of *OsLPR5*, two-week-old wild-type seedlings were treated for 48 h with 120 mM NaCl. The results showed that salt stress triggered a significant induction of *OsLPR5* in the roots and shoots (Figure 1A,B). Furthermore, since *OsLPR5* encodes a ferroxidase, the ferroxidase activity in wild-type seedlings was also measured with or without NaCl treatment, showing that the activity was significantly induced by salt stress (Figure 1C). The above results suggested that *OsLPR5* may play a role in response to salt stress. We examined the expression patterns of *OsLRP5* by qRT-PCR in various wild-type tissues, and it was found that the transcriptional level of *OsLPR5* could be determined both in vegetative and reproductive organs, but was far higher in vegetative organs in the tillering stage (Figure 1D).

### 2.2. Overexpression of OsLPR5 Increased Tolerance to Salinity Stress in Seedling Stage

To investigate the functional role of *OsLPR5* in regulating the salt stress response, we constructed a knock-out mutant by CRISPR/Cas9 and transgenic lines overexpressing *OsLPR5*. Two independent mutants (*lpr5-1* and *lpr5-2*) were generated and their editing sites were identified (Figure 2A). The expression levels of *OsLPR5* in the overexpressing lines (#1, #2 and #3) were increased by more than 1000-fold compared with the wild type, as determined by qRT-PCR (Figure 2B). Two overexpressing lines (#1 and #3) were then selected for further study. Consistently, the ferroxidase activities extracted from the plant total protein in overexpressing seedlings were significantly elevated compared to the WT. In contrast, *lpr5-1* and *lpr5-2* showed lower ferroxidase activities compared to the WT (Figure 2C). Then, the overexpressing seedlings (#1 and #3) were treated with 100, 120, 150 and 200 mM NaCl for 7 days to investigate whether *OsLPR5* is involved in affecting salt tolerance (Appendix A). It was found that OE-*OsLPR5* seedlings suffered less damage, had more green leaves and were less sensitive to salt stress compared with the WT at different levels of salt stress (Appendix A).

OE-*OsLPR5* (#1 and #3), *lpr5-1*, *lpr5-2* and WT seedlings were subsequently stressed with 120 mM NaCl for further phenotypic observation. Notably, the results showed that *lpr5-1* and *lpr5-2* had significantly reduced fresh weights and seedling heights after salt stress compared with the WT (Figure 3A–C). Moreover, both *lpr5-1* and *lpr5-2* contained extremely lower levels of total chlorophyll and much higher levels of an indicator of membrane lipid peroxidation (malondialdehyde (MDA)) under salt treatment, which indicated that the mutation of *OsLPR5* caused an increased sensitivity to salt stress (Figure 3D,E). In contrast, overexpressing *OsLPR5* in rice increased the tolerance to salt stress, with a lower fresh weight loss, seedling height inhibition and MDA content and higher levels of chlorophyll compared with the WT under NaCl treatment (Figure 3A–E). Collectively, the above results suggest that *OsLPR5* regulates the salinity stress tolerance by acting as a new positive regulator in rice.

### 2.3. OsLPR5 May Influence Osmotic Adjustment and ROS-Scavenging Enzyme Activities

Upon exposure to salt stress, plants accumulate compatible osmolytes and promote the biosynthesis of a set of ROS-scavenging systems to resist osmotic and oxidative stress [1]. Therefore, we determined the contents of osmolytes, including soluble protein and proline, in plants. Under normal conditions, there were no significant differences among OE-*OsLPR5* (#1 and #3), *lpr5-1*, *lpr5-2* and WT seedlings. Under NaCl treatment, *lpr5-1* and *lpr5-2* contained markedly lower soluble protein and proline than OE-*OsLPR5* and WT (Figure 4). Moreover, the OE-*OsLPR5* (#1 and #3) seedlings accumulated much higher levels of proline content than the WT (Figure 4). Consistent with this, the ROS scavenging enzyme activities in *lpr5-1* and *lpr5-2*, including superoxide dismutase (SOD), ascorbate peroxidase (APX) and catalase (CAT), were also significantly lower than OE-*OsLPR5* and WT, whereas these enzyme activities were higher in OE-*OsLPR5* compared to *lpr5-1*, *lpr5-2* and WT seedlings (Figure 4). Taken together, OsLPR5 appeared to promote the accumulation of cell osmotic adjustment substances and promote ROS-scavenging enzyme activities in response to osmotic and oxidative stress induced by salt stress in rice.

### 2.4. Mutation of OsLPR5 Mainly Influence Na^+^ Levels under NaCl Stress

Excessive Na^+^ uptake and altered Na^+^/K^+^ homeostasis in shoots lead to leaf damage when exposed to high salt concentrations. Thus, the contents of Na^+^ and K^+^ were measured in the rice shoots with or without NaCl treatment. Under normal conditions, there were no significant differences in Na^+^ content among the plants (Figure 5). The Na^+^ levels of all seedlings tested were dramatically increased after 120 mM NaCl treatment, especially in *lpr5-1* and *lpr5-2* (Figure 5). However, except for the higher K^+^ level of *lpr5-2*, the contents in other plants were not much different, which suggested that OsLPR5 may not directly regulate K^+^ homeostasis. Similar to the Na^+^ levels, the Na^+^/K^+^ ratios also showed a sharp increase under salt stress, especially in *lpr5-1* and *lpr5-2* (Figure 5). These results imply that mutation of OsLPR5 mainly causes excessive Na^+^ levels, Na^+^/K^+^ ratios and salt sensitivity.

### 2.5. OsLPR5 Significantly Changes the Expression Levels of Salt Stress and Iron Homeostasis-Related Genes in Rice

Rice will suffer from osmotic stress, oxidative stress and excessive Na^+^ uptake when exposed to high salt concentrations. Under NaCl treatment, *lpr5* contained markedly lower proline levels and much higher levels of Na^+^ than OE-*OsLPR5* and WT (Figure 4 and Figure 5). Therefore, we evaluated the relative expression levels of the key proline-synthesis-related gene (*OsP5CS1*) and high-affinity sodium transporter (*OsHKT2;1*) in WT, OE-*OsLPR5* and *lpr5* seedlings under normal and NaCl treatment conditions by qRT-PCR [23,24,25]. The results showed that the expression levels of *OsP5CS1* and *OsHKT2;1* were significantly upregulated in WT and OE-*OsLPR5* after NaCl treatment, but downregulated in *lpr5* seedlings (Figure 6). Moreover, the expression of these two genes was most significantly upregulated in the overexpressing seedlings, which is consistent with their increased salt tolerance (Figure 6). Fe is essential for the synthesis of ferredoxins and other redox-related proteins participating in a variety of physiological activities in plants [9,26]. Ai et al. reported that overexpression of *OsLPR5* elevated the concentrations of Fe (III) in the xylem sap and total Fe in rice [22]. In grasses, the acquisition and mobilization of Fe (III) mainly occurs through a chelation-based approach (Strategy II). The YS1-like (YSL) family have been reported to play important roles in this approach. *OsYSL15*, one of the rice YSL genes played a positive role in Fe uptake and distribution during the first stages of growth [27]. Therefore, the relative expression levels of *OsYSL15* was analyzed in WT, OE-*OsLPR5* and *lpr5* seedlings. Under normal conditions, the transcript level in WT, OE-*OsLPR5* and *lpr5* seedlings exhibited no significant difference. However, its level was remarkedly increased in *OsLPR5*-overexpressing seedlings in response to salt stress (Figure 6). We speculated that this was related to the increased ferroxidase activity and concentration of Fe (III) in *OsLPR5*-overexpressing seedlings.

## 3. Discussion

### 3.1. OsLPR5 and Ferroxidase Activities Are Induced under Salt Stress

LPRs are highly expressed in roots and play important roles in root meristem development under low Pi in Arabidopsis (*LPR1* and *LPR2*) and rice (*OsLRP5*) [19,22]. Here, we also used qRT-PCR analyses to determine the expression patterns of *OsLRP5* in different tissues, and found consistent higher expression levels in roots and also in the leaves and stems at the tillering stage (Figure 1D). This may be due to the different tissues and periods of the samples tested in our study. The results suggested the broad spectrum influence of *OsLRP5* on plant growth in rice. Therefore, we investigated the relative transcription levels of *OsLRP5* in roots and shoots under NaCl treatment to determine whether this gene was involved in the salt stress response. It showed that salt stress triggered a significant induction of *OsLPR5* both in the roots and shoots, which suggested its possible role in the regulation of salt stress (Figure 1A,B).

Both LPR1 and OsLRP5 have ferroxidase activity in Arabidopsis and rice by catalyzing the oxidation of Fe (II) to Fe (III) using O_2_ as a substrate [19,20,22]. In Arabidopsis, the ferroxidase activity in root extracts was up to 5-fold higher in *LPR1*-overexpressing lines compared with the wild type (Col) [20]. By detecting the ferroxidase activity of heterologously expressed purified pGS::GST::OsLPR5 fusion proteins in *E. coli*, OsLPR5 also showed ferroxidase activity [22]. In addition, the activity was significantly higher in the roots of *OsLPR5*-overexpressing lines under Pi-sufficient and Pi-deficient conditions [22]. Consistent with this, our study confirmed that the ferroxidase activity was significantly higher in the shoots of *OsLPR5*-overexpressing lines (#1 and #3) than in the WT (Figure 2C). Moreover, the mutants (*lpr5-1* and *lpr5-2*) showed lower ferroxidase activities compared with the WT (Figure 2C). We further measured the ferroxidase activities in WT at the seedling stage with or without NaCl stress and found a significantly increased activity under NaCl stress (Figure 1C). These results indicated that ferroxidase activity was approximately correlated with OsLPR5 transcription levels.

### 3.2. OsLPR5 Positively Regulates the Tolerance to Salinity Stress

Salinity stress can cause various physiological and biochemical changes, including ionic toxicity, osmotic stress, ROS accumulation and nutritional imbalance [2]. Enhanced salt-tolerant plants generally have higher chlorophyll and proline contents, lower MDA content and ROS communication and increased ROS-scavenging enzyme activities, for example, in *OsR3L1* and *OsMADS25* overexpressing seedlings [14,28]. Consistent with this, the *OsLPR5*-overexpressing lines (#1 and #3) had higher fresh weights and chlorophyll contents, lower MDA contents, higher proline contents and increased ROS-scavenging enzyme activities than the WT after NaCl stress. In contrast, the mutants (*lpr5-1* and *lpr5-2*) suffered more severe yellowing, chlorophyll destruction and osmotic stress and had a lower ROS scavenging ability after salt stress (Figure 3 and Figure 4). Consist with this, the transcriptional level of *OsP5CS1* was significantly upregulated in OE-*OsLPR5* after NaCl treatment (Figure 6). These results indicated that OsLPR5 positively regulates the tolerance to salt stress.

An elevated Na^+^/K^+^ ratio is a characteristic manifestation of ionic toxicity with salt stress. For example, there was a higher K^+^/Na^+^ ratio and an enhanced salt tolerance in *OsSTAP1* overexpressing lines compared to the WT [29]. In our study, both the Na^+^ levels and Na^+^/K^+^ ratios were elevated in the mutants (*lpr5-1* and *lpr5-2*) compared with the WT, consistent with the salt-hypersensitivity phenotype (Figure 5). Unexpectedly, overexpression of OsLPR5 did not decrease the Na^+^ level or Na^+^/K^+^ ratio after NaCl stress, as they were approximately the same as those in the WT (Figure 5). We speculated that OsLPR5 confers salt tolerance primarily at other physiological levels. The root system is the first tissue to perceive salt stress, and a greater root volume under salt stress in overexpressing *OsAHL1*, *OsHAL3* and *OsMADS25* seedlings exhibited enhanced salt avoidance [1,30]. Similar to this, the primary root length was longer in OsLPR5-overexpressing lines than the WT at the seedling stage (Appendix A). On the other hand, there have been reports that the expression levels of *OsIRO2*, *OsIRT1*, *OsNAS1*, *OsNAS2*, *OsYSL15* and *OsYSL2* were enhanced by a saline-alkaline environment and the plants could acquire Fe more efficiently, thus contributing to a higher accumulation of Fe [6,31]. The increased concentration of Fe (III) and total Fe in the roots and shoots of *OsLPR5*-overexpressing plants may be one of the reasons for its improved salt tolerance [22]. In addition, the transcriptional levels of *OsHKT2;1* and *OsYSL15* were significantly upregulated in OE-*OsLPR5* after NaCl treatment. We speculated that OsLPR5 may influence the salt stress tolerance by regulating the expressions of *OsHKT2;1* and *OsYSL15*. However, the detailed molecular mechanisms and genetic pathways of OsLPR5-mediated salt tolerance remain to be elucidated.

Taking our current results together with those of previous studies, *OsLPR5* is a newly identified gene belong to *LPRs* that significantly improves the salt stress tolerance in rice. We investigated the plant height and thousand-grain weight of OE-*OsLPR5* in the mature stage and found that overexpressing *OsLPR5* causes a slight decrease in both agronomic traits (Appendix A). However, further work is needed to explore the potential use of OsLPR5 in breeding salt-tolerant rice varieties.

## 4. Materials and Methods

### 4.1. Experimental Materials and Stress Treatment

The rice variety *japonica* cv. Nipponbare was used as the wild type (WT) and genetic transformation in this study. For salt stress, the sterilized seeds were germinated in water at 32 °C and then grown in culture solution (1.15 mM (NH_4_)_2_SO_4_, 0.2 mM NaH_2_PO_4_, 1 mM CaCl_2_, 1 mM MgSO_4_, 0.4 mM K_2_SO_4_, 0.009 mM MnCl_2_, 0.075 mM (NH_4_)_6_Mo_7_O_24_, 0.019 mM H_3_BO_3_, 0.152 mM ZnSO_4_, 0.155 mM CuSO_4_ and 0.02 mM Fe-EDTA, pH 5.8) in a growth chamber at 32 °C light/28 °C dark. The 14-day-old seedlings of WT and transgenic lines were treated with 100, 120, 150 and 200 mM NaCl for one week. Seedlings grown in nutrient solution without NaCl served as controls. After treatment, the fresh weight and shoot length of each plant were measured. Each treatment experiment was performed with three replicates with twelve seedlings. Statistically significant differences (*p* < 0.05 or *p* < 0.01) were identified by an ANOVA followed by Duncan’s test.

### 4.2. RNA Isolation and qRT-PCR

Total RNA was extracted from rice tissues using Trizol (Hlingene, Shanghai, China) according to the manufacturer’s instructions. For RT-qPCR analyses, 1 μg of RNA sample was depleted of genomic DNA and reverse-transcribed into cDNA using Hifair^III^ 1st Strand cDNA Synthesis SuperMix (YEASEN, Shanghai, China). The Hieff qPCR SYBR Green Master Mix premix (YEASEN, Shanghai, China) was used in qRT-PCR experiments. The reaction was performed with biological triplicates as described previously [32]. The transcription levels were calculated by the 2^−∆∆CT^ values with the expression of the *ubiquitin* gene as the internal control to normalize gene expression data. The primers used in this experiment are listed in Appendix A.

### 4.3. Vector Constructs and Rice Genetic Transformation

To construct the OE-*OsLPR5* plasmid, the coding sequence of *OsLPR5* was cloned and constructed into the pCAMBIA1300-UBI-RBCS vector using the Hieff Clone^®^ Plus One Step Cloning Kit (YEASEN, Shanghai, China). For *lpr5* mutant construction, the CRISPR/Cas9 system was used based on a previous report [33]. The specific CRISPR target site was designed in the protein-encodable region by the CRISPR-GE Software Toolkit (http://crispr.hzau.edu.cn/CRISPR2/, accessed on 16 March 2023) [34]. The annealed 19-bp genomic DNA was ligated into pYLgRNA-OsU3 using T4 ligase (New England Biolabs, Shanghai, China). All constructs were transferred into *Agrobacterium tumefaciens* strain EHA105 and genetically transformed into Nipponbare by the Agrobacterium-mediated transformation method to generate transgenic lines following a previous report [35]. The specific primers are listed in Appendix A.

### 4.4. Chlorophyll Content Assay

For total chlorophyll content measurements, 0.1 g of fresh leaves with or without treatment were chopped and put into 10 mL of ethanol/acetone (1:1) mixed solution. Then, pigments were extracted in the dark for 48 h until the leaves turned completely white. Absorbance values at 663 nm and 645 nm were measured with an ultraviolet spectrophotometer. Each sample had three biological replicates. The chlorophyll contents were calculated according to a previously described process [36].

### 4.5. Physiological Measurements

The free proline content in leaves was measured by the sulfosalicylic acid method according to a previous method [37]. Briefly, 0.2 g of leaves from control and treatment plants were homogenized in solution (3% sulfosalicylic acid/glacial acetic acid/2.5% ninhydrin hydrate 1:1:2), followed by inoculation in a 100 °C water bath for 15 min. After cooling, toluene was added and the UV absorption of the toluene layer at 520 nm was measured. The proline content was calculated according to the standard curve. The soluble protein in leaves was homogenized with a Coomassie brilliant blue solution and measured at 595 nm. For MDA content, 0.2 g of leaves was powdered and homogenized in solution (0.25% TBA dissolved in 10% TCA). After centrifugation, the supernatant was inoculated in a 95 °C water bath for 15 min. After cooling, the absorptions at 532 nm, 600 nm and 450 nm were measured [38].

The antioxidant enzyme activities were measured according to a previous method [14,39]. Briefly, fresh leaves from control and treatment plants were ground by liquid nitrogen and then homogenized with potassium phosphate buffer (pH 7.2). For analyses of APX activity, 5 mM vitamin C (AsA) was added and the absorption at 290 nm was measured. The APX activity was determined by the degradation rate of AsA. The total CAT activity was measured by the rate of decomposition of H_2_O_2_ at 240 nm. The SOD activity was assayed by monitoring the percentage of inhibition of the pyrogallol autoxidation at 320 nm. Statistically significant differences (*p* < 0.05) were identified by an ANOVA followed by Duncan’s test.

### 4.6. Quantification of Na^+^ and K^+^ Concentrations

The shoots of rice seedlings from WT and transgenic plants with or without treatment were collected for the determination ion concentrations. All the samples were oven-dried at 105 °C for 30 min and 75 °C for 48 h. The Na^+^ and K^+^ concentrations of each sample were determined according to a previous report [40]. Statistically significant differences (*p* < 0.01) were identified by an ANOVA followed by Duncan’s test.

### 4.7. Determination of Ferroxidase Activity

To determine the ferroxidase activity of plants, total plant proteins were extracted from seedlings of the wild type and transgenic plants as described previously [41]. Samples were ground with liquid nitrogen and immersed in extracted solution containing a Protease inhibitor cocktail. Then, the total plant proteins were obtained by centrifugating twice at 4 °C and collecting the supernatant. For ferroxidase assays, the reaction contained 1050 μL buffer (450 mM Na-acetate (pH 5.8), 100 mM CuSO_4_), 225 μL substrate (357 mM Fe(NH_4_)_2_(SO_4_)_2_·6H_2_O, 100 mM CuSO_4_) and 30 μL plant protein. The solution was mixed gently and 16 μL 18 mM ferrozine was added to quench the rection at regular intervals. The reaction mixture without substrate was used as control. The activity of ferroxidase was assayed by the rate of Fe^2+^ oxidation, which was calculated from the decreased absorbance at 560 nm according to the method in [20,22].

## Figures and Tables

**Figure 1 ijms-24-08115-f001:**
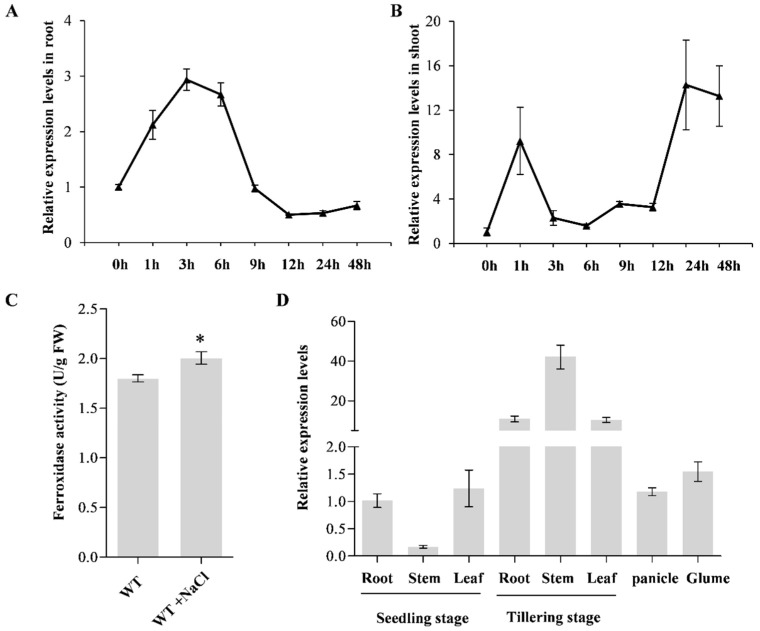
Expression pattern of *OsLPR5*. Time course of *OsLPR5* under salt stress conditions (120 mM NaCl) in roots (**A**) and shoots (**B**) at the seedling stage via qPCR analysis. (**C**) The ferroxidase activity of total protein from wild type (WT) under normal and salt stress conditions. Asterisks represent statistical difference at * *p* < 0.05. (**D**) Expression patterns of *OsLPR5* in different tissues containing root, stem, leaf panicle and glume by RT-qPCR. Data are shown as means ± SD (*n* = 3).

**Figure 2 ijms-24-08115-f002:**
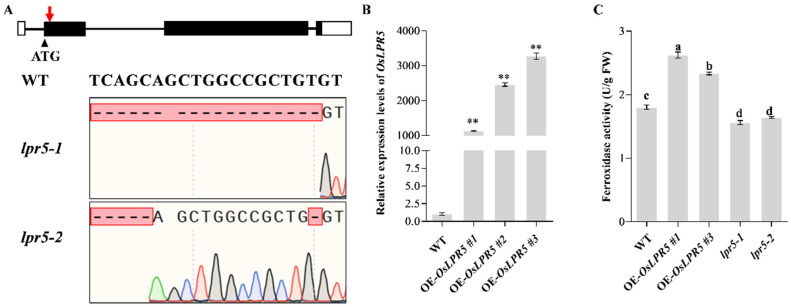
Identification of *lpr5* mutants and overexpressed lines. (**A**) Sequence analysis of *lpr5-1* and *lpr5-2* mutants generated by CRISPR/Cas9. The red arrow indicates the target site. (**B**) Expression analysis of the *OsLPR5* gene in overexpressed transgenic lines. Asterisks represent statistical differences at ** *p* < 0.01. (**C**). The ferroxidase activity assay in WT and *OsLPR5* transgenic seedlings. Equal amounts of total protein from leaves of WT, OE-*OsLPR5* and *lpr5* seedlings were incubated with the substrate Fe(NH_4_)_2_(SO_4_)_2_·6H_2_O. Data are shown as the means ± S.D (*n* = 3). Significant differences (*p* < 0.05) are indicated by different letters.

**Figure 3 ijms-24-08115-f003:**
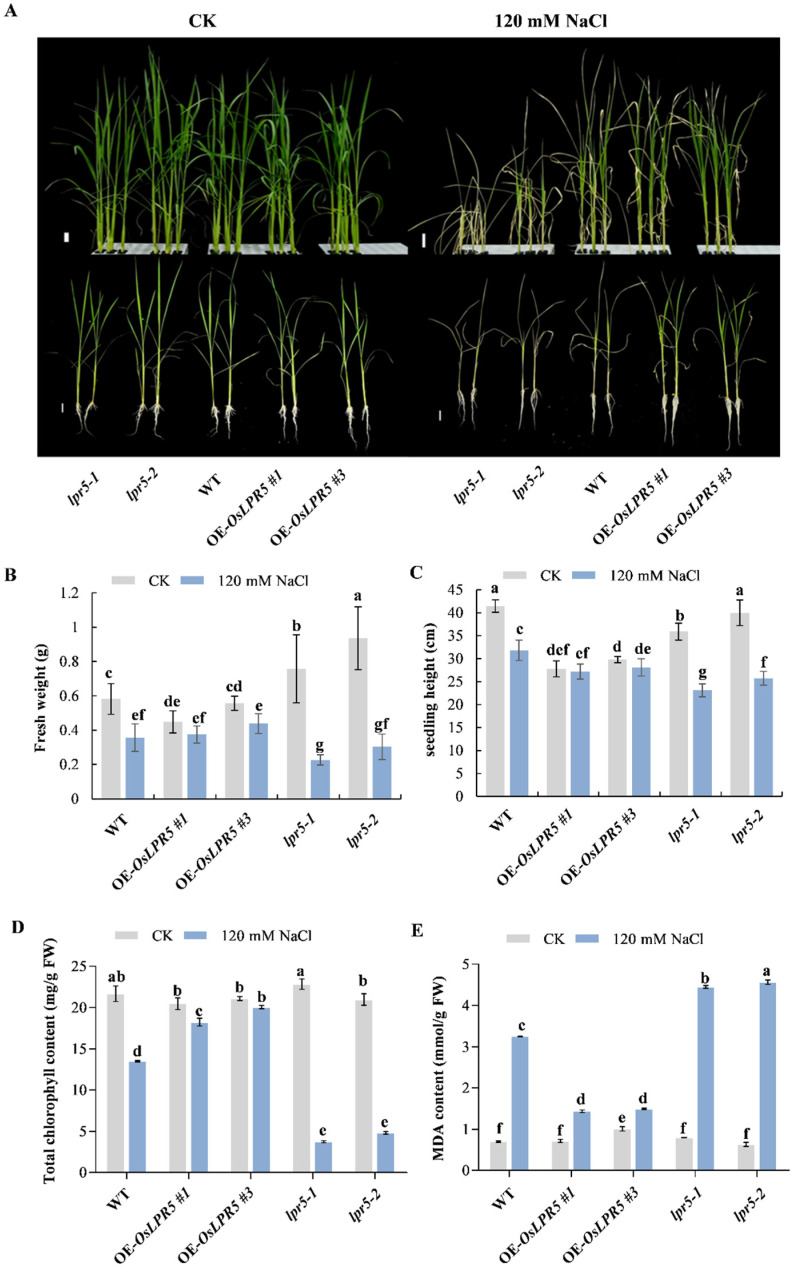
OsLPR5 contributes to the salinity tolerance of rice. (**A**) Phenotype of wild-type and *OsLPR5* transgenic seedlings exposed to salinity stress for 7 days. Scale bars, 2 cm. (**B**–**E**) Measurement of fresh weight (**B**), seedling height (**C**), the total chlorophyll (**D**) and MDA (**E**) contents in wild-type and *OsLPR5* transgenic seedlings exposed to salinity stress for 7 days. Data are shown as means ± S.D (*n* = 3). Different letters indicate significant differences between means as determined using an ANOVA followed by Duncan’s test (*p* < 0.05).

**Figure 4 ijms-24-08115-f004:**
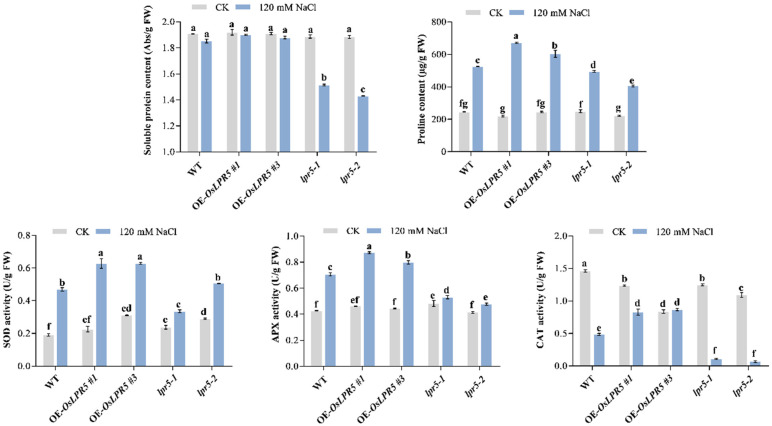
Measurement of the content of soluble protein, proline and ROS-scavenging enzyme activities of SOD, APX and CAT in leaves of wild-type, OE-*OsLPR5* and *lpr5* transgenic seedlings under normal conditions or exposure to salt stress for 7 days. Different letters indicate significant differences between means as determined using an ANOVA followed by Duncan’s test (*p* < 0.05).

**Figure 5 ijms-24-08115-f005:**
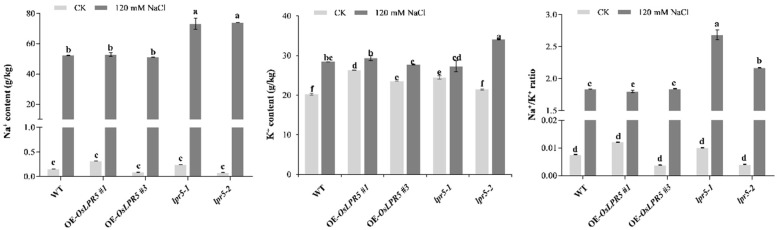
Measurement of the Na^+^ and K^+^ contents and Na^+^/K^+^ ratio in leaves of wild-type, OE-*OsLPR5* and *lpr5* transgenic seedlings under normal conditions or exposure to salt stress for 7 days. Different letters indicate significant differences between means as determined using an ANOVA followed by Duncan’s test (*p* < 0.01).

**Figure 6 ijms-24-08115-f006:**
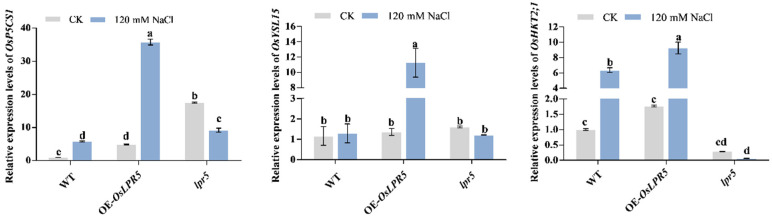
Relative expression levels of stress-related genes in leaves of wild-type, OE-*OsLPR5* and *lpr5* transgenic seedlings under normal conditions or exposure to salt stress for 4 h. Different letters indicate significant differences between means as determined using an ANOVA followed by Duncan’s test (*p* < 0.05).

## Data Availability

The original data of this present study are available from the corresponding authors.

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
