# Peer review of "OsLPR5 Encoding Ferroxidase Positively Regulates the Tolerance to Salt Stress in Rice"

_ijms, 2023, doi:10.3390/ijms24098115_

Round 1

Reviewer 1 Report

Article "OsLPR5 encoding ferroxidase positively regulates the tolerance of salt stress in rice" claims that OsLPR5 is involved in salt tolerance mechanism in rice. They generated over expression and knockout and conducted targeted gene expression and physiological assay.

I have some serious concerns

1. Author conducted Student's t-test. This is inappropriate test. They need to to perform ANOVA + Tukey's multiple comparison test 

2. Most the variation is data seems noisy and it is not enough convincing that overexpression lines are better than wild type under stress condition. They need to do more replications and repeated experiments.

3. More convincing data will be testing in field conditions

4. Expressions pattern in shoot (Figure 1) compared to root seems skeptical. Again authors need to include more replications and repeat experiments from biological independent plants.

Thanks

Author Response

Response to Reviewer 1 Comments

Comments and Suggestions for Authors

Article "OsLPR5 encoding ferroxidase positively regulates the tolerance of salt stress in rice" claims that OsLPR5 is involved in salt tolerance mechanism in rice. They generated over expression and knockout and conducted targeted gene expression and physiological assay. I have some serious concerns.

Point 1: Author conducted Student's t-test. This is inappropriate test. They need to perform ANOVA + Tukey's multiple comparison test.

Response 1: Thank you very much for your efforts toward the improvement of this paper. Please allow me to explain, In Figure 2B, to compare the expression levels of overexpressed lines with WT, the Student's t-test was conducted. And we actually performed ANOVA + Duncan's multiple comparison test by SPSS in Figure 2C and Figure 3-6, but there was a mistake in the manuscript and it has been corrected in the revised version.

Point 2: Most the variation is data seems noisy and it is not enough convincing that overexpression lines are better than wild type under stress condition. They need to do more replications and repeated experiments.

Response 2: as a matter of fact, to confirm that the overexpressed plants enhanced salt tolerance, we repeated salt treatment experiments several times and confirmed this conclusion. According to your suggestions, the repeated experiments was added in supplementary figure 2 in revised version (or you can see below).

Point 3:  More convincing data will be testing in field conditions

Response 3: nice suggestion! However, at present, we do not have the conditions to plant transgenic plants in saline-alkali field. we are sorry that we may not be able to present the data here, but in our future plan, we will screen varieties which have higher expression levels of OsLPR5 and test the impact on salt tolerance of different varieties in saline alkali filed and this will certainly be included.

Point 4: Expressions pattern in shoot (Figure 1) compared to root seems skeptical. Again authors need to include more replications and repeat experiments from biological independent plants.

Response 4: According to your suggestions, the repeated salt stress and qPCR experiments was performed using salt-stressed NIP and it showed similar trends with a slight difference (see below). We believe that it should have not affected the result of this our story.

Reviewer 2 Report

The article presents an interesting study on the role of OsLPR5 in the salt stress response and tolerance in rice. This study contributes to our understanding of the mechanisms underlying salt stress tolerance in rice and provides valuable information for the development of salt-tolerant varieties. However, there are a few potential shortcomings in this article:

Although the article highlights the importance of OsLPR5 in salt stress tolerance in rice, it does not provide a thorough discussion on the potential applications of this knowledge. For example, it would be beneficial to explore the potential use of OsLPR5 in breeding salt-tolerant rice varieties.

The article does not provide a comparison to previous studies that have investigated the role of LPRs in salt stress response and tolerance in plants. It would be useful to see how this study builds on previous knowledge in this field.

The article does not provide a thorough discussion on the potential implications of the research. It would be helpful to explore how this research can inform future studies on the molecular mechanisms underlying salt stress response and tolerance in rice.

Additionally, the article did not report on the long-term effects of OsLPR5 overexpression, and whether it has any negative impacts on the plant's growth and development.

minor comments:

In line 171, it is unclear what "full culture solution" refers to. It would be helpful to specify the components of this solution.

In line 198, "cut into 1 cm pieces" could be replaced with "chopped" to make the language more concise.

Author Response

Response to Reviewer 2 Comments

The article presents an interesting study on the role of OsLPR5 in the salt stress response and tolerance in rice. This study contributes to our understanding of the mechanisms underlying salt stress tolerance in rice and provides valuable information for the development of salt-tolerant varieties. However, there are a few potential shortcomings in this article:

Point 1: Although the article highlights the importance of OsLPR5 in salt stress tolerance in rice, it does not provide a thorough discussion on the potential applications of this knowledge. For example, it would be beneficial to explore the potential use of OsLPR5 in breeding salt-tolerant rice varieties.

Response 1: first of all, the authors would like to show their sincere appreciation to your suggestive comments on this work. We added the thorough discussion according to the reviewer in the last paragraph in the revised version.

Point 2: The article does not provide a comparison to previous studies that have investigated the role of LPRs in salt stress response and tolerance in plants. It would be useful to see how this study builds on previous knowledge in this field.

Response 2: Agree. However, there are five LPRs in rice (OsLPR1-5); OsLPR3 and OsLPR5 revealed strong tissue-specific induction during Pi deficiency. And OsLPR5 was located in ER and cell wall, had the ferroxidase activity, and was required for normal growth and maintenance of phosphate homeostasis in rice. Although important roles of LPRs were reported in plants, their function in salt stress response and tolerance has yet to be defined. We first discovered that OsLPR5 was positively regulating salt-stress tolerance.

Point 3: The article does not provide a thorough discussion on the potential implications of the research. It would be helpful to explore how this research can inform future studies on the molecular mechanisms underlying salt stress response and tolerance in rice.

Response 3: nice suggestion! We added the thorough discussion according to the reviewer in the Discussion in the revised version.

Point 4: Additionally, the article did not report on the long-term effects of OsLPR5 overexpression, and whether it has any negative impacts on the plant's growth and development.

 Response 4: Very constructive suggestions. In the mature stage, OsLPR5-overexpressed plants has shorter plant height and reduced 1000-grain weight compared with wild type (Figure S4 in the revised version).

minor comments:

Point 5: In line 171, it is unclear what "full culture solution" refers to. It would be helpful to specify the components of this solution.

Response 5: It was revised as suggested in the experimental materials and stress treatment method.

Point 6: In line 198, "cut into 1 cm pieces" could be replaced with "chopped" to make the language more concise.

Response 6: corrected.

Round 2

Reviewer 1 Report

Authors addressed my comments and now included more supplementary data that validate the repeatability of experiments.
